# A Novel Spinel Ferrite-Hexagonal Ferrite Composite for Enhanced Magneto-Electric Coupling in a Bilayer with PZT

**DOI:** 10.3390/s23249815

**Published:** 2023-12-14

**Authors:** Sujoy Saha, Sabita Acharya, Maksym Popov, Theodore Sauyet, Jacob Pfund, Rao Bidthanapally, Menka Jain, Michael R. Page, Gopalan Srinivasan

**Affiliations:** 1Department of Physics, Oakland University, Rochester, MI 48309, USA; sujoysaha@oakland.edu (S.S.); sabitaacharya@oakland.edu (S.A.); maxim_popov@univ.kiev.ua (M.P.); burao@oakland.edu (R.B.); 2Institute of High Technologies, Taras Shevchenko National University of Kyiv, 01601 Kyiv, Ukraine; 3Department of Physics, University of Connecticut, Storrs, CT 06269, USA; theodore.sauyet@uconn.edu (T.S.); jacob.pfund@uconn.edu (J.P.); menka.jain@uconn.edu (M.J.); 4Materials and Manufacturing Directorate, Air Force Research Laboratory, Wright-Patterson Air Force Base, Dayton, OH 45433, USA; michael.page.16@us.af.mil

**Keywords:** magnetoelectric, spinel ferrite, hexagonal ferrite, ferroelectric, composite

## Abstract

The magnetoelectric effect (ME) is an important strain mediated-phenomenon in a ferromagnetic-piezoelectric composite for a variety of sensors and signal processing devices. A bias magnetic field, in general, is essential to realize a strong ME coupling in most composites. Magnetic phases with (i) high magnetostriction for strong piezomagnetic coupling and (ii) large anisotropy field that acts as a built-in bias field are preferred so that miniature, ME composite-based devices can operate without the need for an external magnetic field. We are able to realize such a magnetic phase with a composite of (i) barium hexaferrite (BaM) with high magnetocrystalline anisotropy field and (ii) nickel ferrite (NFO) with high magnetostriction. The BNx composites, with (100 − x) wt.% of BaM and x wt.% NFO, for x = 0–100, were prepared. X-ray diffraction analysis shows that the composites did not contain any impurity phases. Scanning electron microscopy images revealed that, with an increase in NFO content, hexagonal BaM grains become prominent, leading to a large anisotropy field. The room temperature saturation magnetization showed a general increase with increasing BaM content in the composites. NFO rich composites with x ≥ 60 were found to have a large magnetostriction value of around −23 ppm, comparable to pure NFO. The anisotropy field H_A_ of the composites, determined from magnetization and ferromagnetic resonance (FMR) measurements, increased with increasing NFO content and reached a maximum of 7.77 kOe for x = 75. The BNx composite was cut into rectangular platelets and bonded with PZT to form the bilayers. ME voltage coefficient (MEVC) measurements at low frequencies and at mechanical resonance showed strong coupling at zero bias for samples with x ≥ 33. This large in-plane H_A_ acted as a built-in field for strong ME effects under zero external bias in the bilayers. The highest zero-bias MEVC of ~22 mV/cm Oe was obtained for BN75-PZT bilayers wherein BN75 also has the highest H_A_. The Bilayer of BN95-PZT showed a maximum MEVC ~992 mV/cm Oe at electromechanical resonance at 59 kHz. The use of hexaferrite–spinel ferrite composite to achieve strong zero-bias ME coupling in bilayers with PZT is significant for applications related to energy harvesting, sensors, and high frequency devices.

## 1. Introduction

Multiferroic materials exhibit more than one ferroic order, such as ferromagnetism, ferroelectricity, and ferroelasticity [1]. They have recently attracted significant attention due to their potential for applications in spintronics, magneto-electrics, nonvolatile memories, sensors, and electrically tunable magnetic microwave devices [2,3]. A ferromagnetic–ferroelectric composite is a multiferroic that shows a variation of its ferroelectric order parameters when subjected to an external magnetic field, direct magnetoelectric (ME) effect, or changes in magnetic parameters in an applied electric field, converse ME effect [4]. In ME materials, the induced electric polarization P is related to the applied external magnetic field H by P = αH, where α is a second order ME-susceptibility tensor. Another parameter of importance is the ME voltage coefficient (MEVC) α_E_ = δE/δH, where δE is the induced electric field due to applied magnetic field δH, and is related to α by α = ε_0_ε_r_α_E_, where ε_r_ is the relative permittivity of the material. According to models for the ME effects in single-phase materials, the upper bound α is limited to the relation α ≤ (με)^1/2^, where μ and ε are the permeability and permittivity of the material, respectively [5]. One known single-phase multiferroic with a large α at room temperature is BiFeO_3_ [6]. In composite ME materials, α can be enhanced by exploiting the strain mediated interactions between the two phases [4].

To obtain the maximum ME voltage coefficient (α_E_) in a ferromagnetic–ferroelectric composite, an optimized magnitude of the DC magnetic bias field H is needed. A maximum in the α_E_ occurs when the piezomagnetic coefficient q = dλ/dH (where λ is the magnetostriction) of ferro/ferrimagnetic component of the composite is also maximum. Hence, a bias magnetic field is generally essential to achieve a strong ME response. The need for a bias field, however, could be eliminated with a self-bias in the ferromagnetic. There are several avenues to accomplish the self-bias condition, such as a large magneto-crystalline anisotropy field or the use of a functionally graded ferromagnet [7,8,9]. Other avenues include the use of a compositionally graded ferromagnetic phase [10]. There are several experimental findings [11,12,13] and theoretical models [11,12,13] on graded ME composites. In laminated composites, changing the mechanical resonance modes through electrical connectivity evokes zero bias coupling when the bending strain activates a built-in bias [9]. Thin films that rely on magnetic field dependence of resonant frequency and angular dependence of exchange bias field [14,15] can also show a zero-bias ME effect. It is also shown that a homogeneous magnetostrictive phase can also produce a zero bias ME effect [16]. Cofired layered composites consisting of textured Pb(Mg_1/3_Nb_2/3_)O_3_-PbTiO_3_ (PMN-PT) and Cu and Zn doped NiFe_2_O_4_ show a giant zero-bias ME coefficient ~1000 mV/cm Oe [17], wherein built in stress induces zero-bias effect. Very recently, Huang et. al. [18] have shown a very large self-bias ME effect with LiNbO_3_ single crystal and Ni trilayers with residual stress engineering by using Ni as the electrode and the piezomagnetic layer simultaneously using RF sputtering. Wu et. al. [19] have shown that large sensitivity in PMN-PT-Metglass ME sensors by utilizing shear stress induced self-bias effect. Annapureddy et. al. [20] have obtained a large self-bias effect (~4200 mV/cm Oe) by utilizing the graded magnetization. To date, all the reported self-bias effects are sample specific. The role of the sample configuration and/or preparation is crucial to obtain the self-bias.

This work focuses on a novel, never-before-used approach for a self-biased ME composite with the use of a ferromagnetic layer consisting of both M-type barium ferrite hexagonal ferrite, BaO 6Fe_2_O_3_ (BaM), with uniaxial anisotropy on the order of ~17.4 kOe, and nickel ferrite NiFe_2_O_4_ with high magnetostriction and piezomagnetic coefficient q [21]. Composites of BaM-NFO with (100 − x) wt.% of BaM and x wt.% of NFO, (BNx), were prepared by sintering powders of both ferrites. X-ray diffraction revealed the presence of both BaM and NFO and the absence of impurity phases. Scanning electron microscopy images showed crystallites of both ferrites. Magnetization measurements at room temperature for static fields up to 2 T showed an increase in 4πM with increasing BaM content in the composite. Magnetostriction λ measurements for BNx indicated an increase with increasing x, and for x > 65, reached values comparable that of pure NFO. High frequency measurements were carried out to determine the anisotropy field H_A_ from dependence of the ferromagnetic resonance (FMR) frequency f_r_ on static magnetic field H. The in-plane H_A_ values in BNx were found to be well above the value of 500 Oe for pure NFO. These composites show a large piezomagnetic coefficient and large magnetocrystalline anisotropy simultaneously, which is somewhat unique to this spinel ferrite–hexaferrite composite. Platelets of sintered BNx were bonded to PZT to form bilayers for ME voltage coefficient (MEVC) measurements that showed a significant zero-field ME effects. Details on results of the studies are provided in the following sections. 

## 2. Experiment

### 2.1. Materials

The ferrite composites were prepared by the traditional ceramic synthesis techniques. Micrometer-sized polycrystalline NFO and BaM powders were first synthesized separately. High purity NiO, BaCO_3_, and Fe_2_O_3_ were mixed and ball milled for 8 h. The powders were dried and pre-sintered at 900 °C for 6 h. The pre-sintered powders were ball milled again, and then sintered at 1200 °C for 6 h. Composites of (100 − x) wt.% BaM-x wt% NFO (BNx) (x = 5, 9, 13, 33, 38, 41, 44, 47, 60, 75, 85 and 95) were prepared by mixing the ferrite powders. A binder, 2% PVA, was added to the powder and pressed into disks (diameter ~18 mm and thickness ~2 mm) by applying uniaxial pressure of 250 MPa. The disks of BNx were finally sintered at 1250 °C for 6 h.

### 2.2. Characterizations

The crystal structure of the composites was characterized by a powder X-ray diffractometer (Miniflex, Rigaku, Japan) at room temperature. Morphological features of the samples were studied with an SEM (JSM-6510/GS, JEOL, Tokyo, Japan). Magnetostriction of the composite on rectangular platelets was measured using a strain gauge and a strain indicator/recorder (P3, Micro-Measurements, Wendell, NC, USA). The magnetic field was applied parallel to the sample plane and along the length (direction-1) of the gauge, and magnetostriction measured in this configuration is labeled λ_11_. Magnetization at room temperature as a function of H was measured with an Evercool Physical Property Measurement System (PPMS, Quantum Design Inc., San Diego, CA, USA). Ferromagnetic resonance (FMR) measurements were conducted on thin rectangular platelets of the composites with the sample placed on a coplanar waveguide and with the magnetic field H parallel to the sample plane and along its length. A vector network analyzer (E8361A, Agilent, Santa Clara, CA, USA) was used to record profiles of the scattering matrix S_21_ vs. frequency f for a series of H. Measurements of ME coupling strengths were completed on a bilayer of BNx and vendor-supplied PZT that was bonded with a thin layer of epoxy. Composite platelets of dimensions 10 mm × 5 mm and 1 to 1.5 mm in thickness, and 0.3 mm thick PZT plates of similar lateral dimensions as the composite, were used. The ME voltage coefficient (MEVC) measurements were completed by subjecting the bilayer to an ac magnetic field H_ac_ and a bias field H. Both H- and f-dependence of the MEVC were measured for both the magnetic fields parallel each other and either parallel or perpendicular to the sample plane. 

## 3. Results

### 3.1. Structural Characterization

Representative powder X-ray diffraction (XRD) patterns for the BNx samples are shown in Figure 1. XRD patterns of other BNx compositions are shown in the Appendix A in the Appendix A. The XRD patterns show diffraction peaks from NFO and BaM. With increasing x-values, the NFO lines become stronger and BaM lines get weaker, as expected. This is due to the reduced weight fraction of the BaM phase as we increase x. X-ray intensity corresponding to a particular phase is proportional to the weight fraction of the phase. We have deliberately varied the weight fraction of the NFO/BaM phases, and when the NFO weight fraction is increased, the line intensity corresponding to NFO becomes stronger; likewise, BaM intensity gets weaker. A small amount of an impurity phase, identified as antiferromagnetic Ba_5_Fe_2_O_8_, is present only in BN41, BN44, and BN75 [22]. Usually, Ba_5_Fe_2_O_8_ type impurities (5BaO + Fe_2_O_3_) occur during the sintering of BaO and Fe_2_O_3_ based ferrites, especially in BaFe_12_O_19_ [22]. However, since Ba_5_Fe_2_O_8_ is antiferromagnetic, the overall ME character of the BNx-PZT bilayers is expected to be unaffected.

Representative SEM images for BNx (x = 5, 33, 60 and 95) are shown in Figure 2. The gradual increase in the grain size in the BN composites with increasing x (also in Appendix A) may indicate that NFO aids in the growth of hexagonal BaM grains for x ≥ 9. Large grains are absent in BN5, but a closer examination of the surface morphology shows hexagonal-like features with grain size less than 2 μm. With increasing NFO content, grains larger than 5 μm are present. For x > 41, the number of large grains reduces again. For the highest content of NFO (BN95), we observe a similar grain distribution as pure NFO (Appendix A). BN5 also shows a similar grain distribution as pure BaM. When the weight fraction of the BaM phase is high (x~5), the composite is more likely to behave as pure BaM. Hence, the corresponding morphological features of composites with high BaM content should also look like pure BaM. Similarly, when the weight fraction (x > 41) of NFO is increased, the composites tend to show a reduction in BaM grains. Barium ferrite itself tends to form hexagonal grains [23] and, with a tweak in the synthesis process, the particle shape can be made nearly spherical [24]. In our case, the processing has a large impact on the grain growth on these composites. The stabilization of the typical hexagonal BaM grains amongst NFO particles is worth noting.

The XRD results in Figure 1 and the SEM images of Figure 2 are indicators of the absence of any significant amount of impurity phases of crystal structures apart from spinel and hexagonal phases. An in-depth investigation and analysis of the crystal and magnetic structures are in order. For example, possible migration of Ni-ion from the spinel phase to the hexagonal phase has to be addressed since the M-type hexagonal phase also has the spinel blocks. Such an investigation, however, is not the primary focus of this particular study.

### 3.2. Magnetic Characterization

We have carried out room-temperature measurements of the magnetization, 4πM, of the composites as a function of applied magnetic field H. Representative 4πM vs. H data are shown in Figure 3 and Appendix A. The M vs. H loops show hysteresis and remanence as expected, and the saturation values of M increases with increasing BaM contents in the composites. The H-values for saturation of the magnetization is less than 3 kOe for NFO rich composites, and it increase as the amount of BaM increases. The magnetization 4πM at H = 20 kOe increases from 2.90 kG for BN95 to 4 kG for BN33. The highest value of remanent 4πM of 1.04 kG was measured for BN75. 

Since the magnetostriction λ is one of the key parameters that determine the strength ME interaction, we measured its value for the BNx composites. The measurements were completed with a strain gauge and a strain indicator and, for H, were was applied parallel to the sample plane and to the length of the strain gauge. Data on the magnetostriction λ_11_ vs. H are shown in Figure 4. The samples exhibit negative values for λ_11_ as expected since both NFO and BaM have negative λ_11_ values [25,26]. None of the λ_11_ values for the composites in Figure 3 show saturation for the maximum H-values of ~3 kOe. With increases in the NFO content, λ_11_ values at 3 kOe increase and tend to show similar behavior as pure NFO (shown in the inset of Figure 4). The highest value of λ_11_ ~−23 ppm was measured for BN60, and similar values were obtained for x ≥ 60. Our measurements of pure BaM showed λ_11_ = −1 ppm at 2.7 kOe. Hence, it is clear that the NFO phase is the major contributor to the net magnetostriction of BNx composites. 

### 3.3. Ferromagnetic Resonance

A key objective of this work is to achieve large enough magneto-crystalline anisotropy field H_A_ in BNx to realize a strong zero-bias ME effect in a composite with PZT. The magnetization data in Figure 3 and Appendix A indicate a large remnant magnetization, as high as 1 kG, which is clear evidence for a large H_A_. We utilized ferromagnetic resonance (FMR) studies in combination with the magnetization data to determine H_A_. Ferrite platelets, rectangular in shape, were placed in an S-shaped coplanar waveguide and excited with microwave power from a VNA. Profiles of the scattering matrix S_21_ as a function of frequency f were recorded. Figure 5 shows such profiles for a series of in-plane H along the sample length. For x values < 10, a single resonance mode was seen in the 50 GHz range (See Appendix A). As NFO content increased, two resonance modes were seen as in Figure 5, one in the frequency range 3–20 GHz and another in the range 40–60 GHz. The S_21_ vs. f profiles in Figure 5 (and Appendix A) show clear asymmetry in the shape of the resonance absorption signals which can be attributed to the variations in the magnitudes of coupling between resonator and the transmission line at frequencies below and above the resonance. Such an asymmetry is not generally observed in cavity-type FMR measurements at a fixed frequency. Also, this effect is negligible for resonance modes with relatively narrow linewidth. However, in the case of transmission line broadband measurement systems and resonances with frequency-width of the order of a few GHz, asymmetry may manifest. Another possible factor is the frequency-dependent background absorption of the coplanar line, which superimposes on absorption through the resonator and leads to significant distortion of the resultant profile. Such asymmetry is most likely to occur at U-band frequencies, where any imperfections of the stripline, connectors, or shielding may unpredictably affect the shape of stripline transmission characteristics. The resonance frequency was estimated from frequency of maximum absorption in the profiles in Figure 5. As discussed, next, the resonance mode at the low frequency region in Figure 5 is due to FMR in NFO whereas the higher frequency resonance is a magneto-dielectric mode in the composite [27].

The H-dependence of the low- and high frequency mode frequencies f_r_ are shown in Figure 6 (and in Appendix A). Data on f_r_ vs. H for the low-frequency mode are shown in Figure 6a for BN33 and BN60. With f_r_ increasing from 8.6 GHz at H = 1 kOe to 15.9 GHz for H = 3.5 kOe for BN60, which amounts to an increase in f_r_ at the rate 2.8 GHz/kOe. A similar rapid increase in f_r_ with H is seen in Figure 6a for BN33. Based on the rate of change in f_r_ with H, one may associate this mode with FMR in NFO. Figure 6b shows f_r_ vs. H for the high frequency mode for BN33 and BN60. In Figure 5 and Appendix A, this mode shows a variation in f_r_ with H that could be approximated to a linear increase ~1.3 GHz/kOe. This slow variation in f_r_ with H is indicative of a magneto-dielectric mode in the composite platelet. This mode is not of importance for the current study and is not considered in further analysis [27].

The built-in bias due to magnetic anisotropy field H_A_ in the composites is a key parameter that accounts for the zero-bias ME coupling in bilayers of the ferrite composites with PZT, as discussed later. There are several avenues for the determination of H_A_, such as direct measurements and from M vs. H data. We utilized FMR results in combination with magnetization values to estimate H_A_. Data on f_r_ vs. H in Figure 7 (and Appendix A) were fitted to appropriate expression for f_r_ to determine the effective magnetization 4πM_eff_ = 4πM + H_A_, where 4πM is the magnetization and H_A_ is the magnetocrystalline anisotropy field. It is essential to note that 4πM is not the saturation magnetization since the M vs. H in Figure 3 (and Appendix A) clearly indicate that there is no saturation of M with H for several of the composites used in this study. We instead used the average value of 4πM for H-values for FMR profiles in Figure 5. 

The resonance frequency f_r_ for FMR mode is given by the Kittel equation,
(1)fr=γH+Nz−Nx4πMeffH+Ny−Nx4πMeff12
where γ is the gyromagnetic ratio, H is the in-plane external magnetic field along the x-direction, N_x_, N_y_, and N_z_ are the demagnetization factors along the length, width, and thickness of the platelet, respectively. These values are shown in Table 1 for each of the composites in which FMR mode was observed. The data in Figure 7 (and Appendix A) were fitted to Equation (1) to determine 4πM_eff_. The Kittel equation in the presented form is, strictly speaking, applicable to the ferromagnetic samples of ellipsoidal shape, magnetized to saturation, and has uniform static and dynamic magnetization. However, the samples presented in this investigation had a parallelepiped shape rather than an ellipsoidal one. That means that, even after an external magnetic field H > N_x_4πM was applied and domain structure was suppressed, the sample was still not in a uniformly magnetized state. There were regions of ferrite present (mostly around edges and corners [28]) where the magnetization deviates from the direction of bias magnetic field. Thus, an even larger H should be applied before the magnetic state of the sample becomes uniform and the Kittel equation becomes applicable. For these reasons, we took only the high-frequency (and high-field) portion of the dependencies shown in Figure 7 and fit them with the Kittel equation to obtain the most reliable fitting parameters.

Estimated values of the gyromagnetic ratio γ and 4πM_eff_ from the fits and the average values of 4πM for H = 1 kOe to 3.5 kOe (from M vs. H data) are given in Table 2. The values of γ range from 3.17 GHz/kOe for x = 33 to 2.61 GHz/kOe for x = 0.60, which are comparable to the 3.0 to 3.2 GHz/kOe value reported for pure NFO [29,30]. The value of the anisotropy H_A_ estimated from FMR data is also given in Table 2, and will have an error of at least 0.5 kOe. The anisotropy field H_A_ is positive for x = 33–95, indicative of in-plane anisotropy for all the polycrystalline BNx composites with x = 33–95. The anisotropy increases from ~0.90 kOe for x = 33 to ~7.77 kOe for x = 75. A further increase in x results in a decrease in H_A_, but the in-plane character of the anisotropy remains. One may therefore infer from these H_A_ -values that a majority of BaM crystallites in x = 33–95 have a unique in-plane orientation leading to positive values of magnetic anisotropy. With increasing value of NFO content in x ≥ 33, the higher concentration of NFO appears to promote the growth of BaM crystallites with in-plane orientation for the c-axis and a net in-plane anisotropy field that reaches a maximum value for x = 75. Several reported efforts in the past on polycrystalline BaM mainly dealt with textured thin and thick films and showed, depending on the degree of texture, an out-of-plane H_A_ in the range 4.5 to 15 kOe [31,32]. In this work, however, the BaM component in composites results in overall effective in-plane anisotropy field.

We have also calculated the Gilbert damping coefficient (GDC) [33,34] of the composites by analyzing the FMR spectra of the samples. GDC is a dimensionless quantity which can be used as a measure of the losses in a ferromagnetic material. GDCs of the composites were calculated using the equation,
(2)$$\alpha =\frac{\gamma \Delta H}{4\pi f_{r}}$$
where α is the damping coefficient, f_r_ is the resonance frequency, ΔH is the linewidth that was estimated from the FMR frequency-width for profiles in Figure 5, and γ is the gyromagnetic ratio. Estimated values are given in Table 2. Composites with x ≤ 47 show GDC ~0.024, and we get a smaller GDC of less than 0.02 for x ≥ 60, which is indicative of a decrease in the losses in the composites as the NFO increases. The composites seem to have a much larger damping coefficient compared to pure and doped NFO [34,35], but it is smaller than the GDC of BaM [36].

### 3.4. Magneto-Electric Effects in the BNx-PZT Bilayers

The strength of ME coupling was measured in bilayers of the composites and vendor-supplied PZT (PZT850, American Piezo Ceramics, Mackeyville, PA, USA). Ferrite platelets of approximate lateral dimensions 5 mm × 10 mm and thickness t = 0.3–0.5 mm were bonded to PZT with 20 μm thick layer of a fast-dry epoxy. The ME voltage coefficient (MEVC) was measured for two different orientations of the applied magnetic fields: (i) α_31_ for the DC filed H and ac field H_ac_ were parallel to each other and along the length of the sample (direction-1), and the induced voltage was measured across the thickness of PZT (direction-3) and (ii) α_33_ for the magnetic fields was applied perpendicular to the sample plane (direction-3) and induced voltage was measured across PZT thickness. The MEVC is given by α = V_3_/(H_ac_ t), where V_3_ is the strain-induced ac voltage measured across PZT. The ME coefficients were measured at low frequencies, as well as at the mechanical resonances in the bilayers.

Figure 8 (and Appendix A) shows the H dependence of α_31_ for ac field at 100 Hz. The MEVC is directly proportional to the piezomagnetic coupling q = dλ/dH. The value of α_31_ increases when H is increased to the maximum. The maximum value of MEVC for BN5 is rather small due to low q-value for this BaM rich composite. With further increases in the NFO content in the composites, α_31_ increases to a maximum value of ~152 mV/cm Oe for BN95. Upon further increase in H, α_31_ in general decreases to a minimum for composites with x > 5. Bias field H dependence of α_31_ in Figure 8 essentially tracks the variation in q with H, reaches a maximum at the maximum in the slope of λ_11_ vs. H, and drops to near zero when the magnetostriction (Figure 3) shows near saturation. Other significant features in the results from Figure 8 are as follows. (i) When H is decreased from ~3 kOe back to zero a hysteresis in α_31_ vs. H is evident for all the BNx-PZT bilayers. (ii) Upon reversing the field H, a 180 deg phase shift (indicated by negative values for α_31_) is observed in the ME voltage except for x = 5. (iii) The bilayers show a finite remnant value for α_31_ at zero-bias that is, as discussed later, attributed to the built-in bias in BNx provided by the anisotropy field H_A_.

Figure 9 shows results of MEVC measurements for the magnetic fields applied perpendicular to the bilayers of BNx-PZT (also shown in Appendix A). The variation in MEVC has similar hysteresis and remanence as the in-plane magnetic fields. However, the following features in α_33_ vs. H differ from the dependence of MEVC for in-plane magnetic fields. (i) Overall MEVC values are smaller in Figure 9 since demagnetization factors reduce both the dc and ac magnetic fields. (ii) The decrease in MEVC for H > 1.5 kOe is relatively small compared to α_31_ vs. H. (iii) Data in Figure 9 for BN95-PZT bilayers show a reversal in the sign of for H > 0.75 kOe for both positive and negative H.

The strength of ME coupling in the bilayers was also characterized by measuring the ac magnetic field frequency f dependence of the MEVC at mechanical resonance modes. Prior to these measurements, we obtained the electromechanical resonance (EMR) frequencies (f_r_) for the composites by measuring the frequency dependence of the impedance with an LCR meter. Mode frequencies could not be obtained for BNx for x ≤ 19, but we were able to determine the frequency of longitudinal resonance modes for higher x-values. MEVC α_31_vs f under a bias field H = 100–200 Oe for the bilayers is shown in Figure 10. One observes an increase in α_31_ with increasing f and a sharp peak in its value at f_r_ ~ 55–75 kHz. We were able to identify f_r_ with the longitudinal EMR mode from the known composite dimensions. For x = 33, the figure shows a fine structure with a double peak in the α_31_vs f profile with values of 147 mV/cm Oe at 72.9 kHz and 132 mV/cm Oe at 73.2 kHz. Bilayers of BNx-PZT with x > 33 have a single peak in the profiles and BN95-PZT shows the highest value of α_31_ = 992 mVcm^−1^Oe^−1^ at 59 kHz. One observes a significant enhancement in the ME coefficients at f_r_ compared to values at 100 Hz (Figure 8), and for example, by a factor 36 for x = 41. The highest Q-factor obtained from this data is found to be 130.4 for BN44. BN41 also has a large Q-factor of 118. After BN44, the Q-factor reduced to 75 at BN60. We discuss the results of these ME measurements in the following section.

## 4. Discussion

It is evident from the results of this study that (i) it is possible to synthesize composites of spinel and M-type hexagonal ferrites free of ferromagnetic impurity phases, (ii) the composites, depending on the amount of BaM, have a moderately high induced planar anisotropy in all compositions, and (iii) the magnetostriction is quite small for BaM rich composites but increases significantly with increasing NFO content; however, the piezomagnetic coefficient q is rather small in all of the composites due to a slow increase in λ with H compared to pure NFO.

It is interesting to note that the coercive field estimated from M vs. H data for the BNx composites remains well under 0.3 kOe for all compositions from x = 33–95. The coercive field gradually increases from 35 Oe for BN33 to 256 Oe for BN95. BN75, the material that has highest anisotropy field of 7.77 kOe, has a coercive field of ~45 Oe. It is also important to note that the highest remanent magnetization obtained for BN75 also has the highest anisotropy. The anisotropy acts as a driving force that gives rise to a remanent magnetization for all BNx samples and a zero-bias MEVC in bilayers with PZT. All the BNx samples remain soft magnetic material with a coercive field less than 300 Oe with no significant hysteresis loss for x = 33–95. The value of this coercive field is well below the coercive field of pure polycrystalline BaM which possesses a large coercive field ~5 kOe. In the composites, the stabilization of the BaM grains (Figure 2) does not seem to increase the coercive field.

It is also clear from the results of ME measurements in Figure 8, Figure 9 and Figure 10 that the bilayers of BNx and PZT show MEVCs that are much higher than the values reported for M-type hexaferrite-PZT bilayers [37], but smaller than those reported for NFO-PZT [38]. Under the optimum value of H, the highest MEVCs are α_31_ = 152 mV/cm Oe and α_33_ = 90 mV/cm Oe, both for BN95-PZT. These values, however, are relatively small due to the weak piezomagnetic coupling strengths in BNx compared to nickel ferrite or nickel zinc ferrite-based layered composites with PZT [34,39].

A key and primary objective of this work was to synthesize a ferromagnetic oxide with a moderately large H_A_ and high magnetostriction and piezomagnetic coupling for use with PZT to achieve ME coupling in the absence of an external bias magnetic field. It is worth noting that this goal was indeed accomplished. Bilayers of BNx-PZT used in this study do show a zero-bias MEVC (Appendix A). Bilayer of BN75-PZT shows the highest α_31_ = 22 mV/cm Oe at zero-bias, and the BN85-PZT bilayer shows the highest α_33_ = 9 mV/cm Oe at zero-bias. It is noteworthy that the remanent magnetization of 1.04 kG for BN75 is the highest for the composites (Table 2). Strategies employed in the past to realize zero-bias ME effects included the use of an external stimuli or functionally graded composites, either in magnetization or in composition, etc. [7,8,9,10,11,12,13,14,15,16,17]. The use of an easy to synthesize composite of a spinel ferrite and hexaferrite in this work for zero-bias ME effect makes this method more viable than others. There are reports wherein composites consisting of NFO and PZT show a large ME coefficient of 460 mV/cm Oe for bilayers and ~1200 mV/cm Oe for multilayers [40]. The MEVC at resonance in these systems was as high as ~1 V/cm Oe [41]. Modified NFO and PZT multilayers even showed a higher ME coefficient [42]. However, there is hardly any evidence for ME coupling zero-bias effect in these composites [37,38,39,40,41,42,43].

Due to very low magnetostriction, BaM is not suitable for strong direct ME coupling, but the very high uniaxial anisotropic field in the system was utilized in this work. Even though BaM grains in our BNx composites are expected to be completely randomized, leading to a net zero the anisotropic field, the increase in the NFO content in BNx seems to promote the growth BaM grains with in-plane c-axis and a net in-plane anisotropy field.

Finally, we compared the zero-bias MEVC values with results reported in the past. Use of a nickel zinc ferrite graded either in magnetization or composition in a bilayer with PZT resulted in a zero-bias MEVC of 37 mV/cm Oe. Electric field-induced bending vibration mode generated zero-bias ME effect in lead free system, shows an MEVC ~30 mV/cm Oe [9]. The low field hysteresis-based zero-bias effect also showed a value of ~60 mV/cm Oe [16]. In our work, we have obtained a zero-bias ME coefficient ~22 mV/cm Oe for BN75-PZT bilayer, which is comparable to the earlier report [10]. The zero-bias ME response in our study could be improved with the use of composites of NFO and M-type strontium ferrite (SrM) or Al substituted SrM or BaM with higher λ than pure BaM. Substituted BaM or SrM may be good choices as they also have anisotropic fields as high as ~30 kG [43].

## 5. Conclusions

In this work, we have successfully synthesized a novel ferrimagnetic composite consisting of (i) nickel ferrite with high magnetostriction and (ii) M-type barium hexaferrite with very high magneto-crystalline anisotropy field. The aim was to use such a high-q and high-H_A_ composite to achieve strong ME coupling in the absence of a bias magnetic field in a bilayer with PZT. BNx composites with x = 5–95 wt.% had high q for NFO rich compositions and in-plane H_A_ as high as 7.77 kOe for x = 75. ME voltage coefficient measurements at low frequencies and at resonance modes showed moderately strong ME coupling at zero bias for samples with NFO content ≥33 wt.%. The highest zero bias MEVC of 21.82 mVcm^−1^Oe^−1^ was obtained for BN75-PZT bilayers wherein BN75 also possesses the highest anisotropy. BN41-PZT shows MEVC ~800 mVcm^−1^Oe^−1^ at electromechanical resonance at 68.4 kHz. The BNx-PZT composites have the potential for use in energy harvesting and sensor technologies.

## Figures and Tables

**Figure 1 sensors-23-09815-f001:**
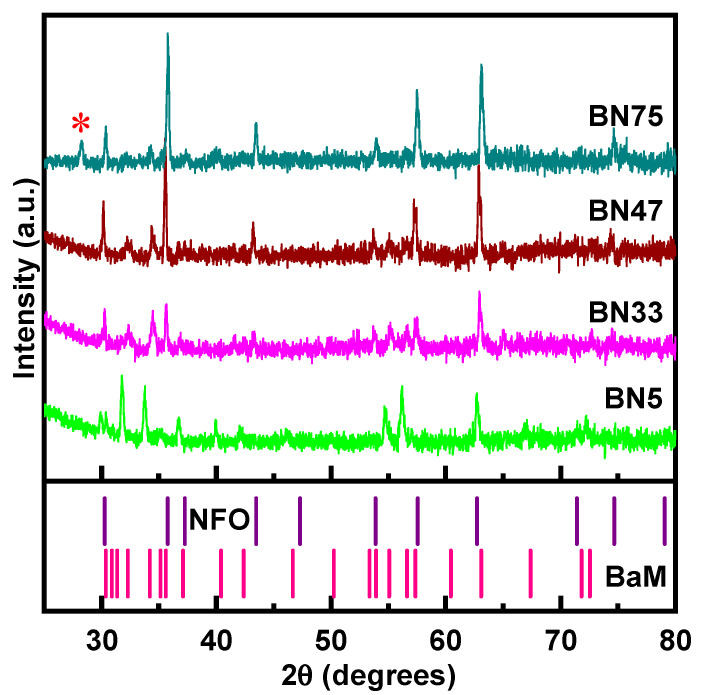
Representative X-ray diffraction data for BNx composites. All the composites bear the signatures of NFO and BaM. The stick patterns for NFO and BaM are shown in the bottom pane to visualize the one-to-one correspondence of the Braggs positions of each phase to the respective NFO and BaM lines. BN75 contains a small amount of an impurity phase, Ba_5_Fe_2_O_8_, and is denoted by a star.

**Figure 2 sensors-23-09815-f002:**
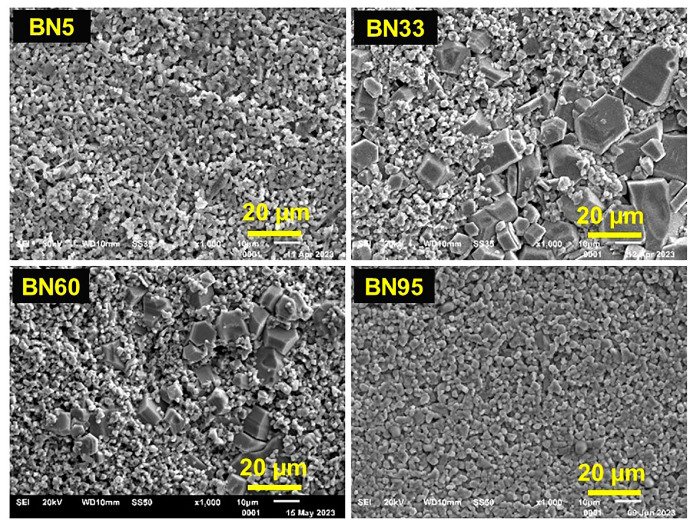
SEM images for BN5, BN33, BN60, and BN95 composites. There are no notable features in the image for BN5. Well-defined hexagonal grains corresponding to the BaM phase are seen in BN33 and BN60. BN95 does not show any hexagonal grain.

**Figure 3 sensors-23-09815-f003:**
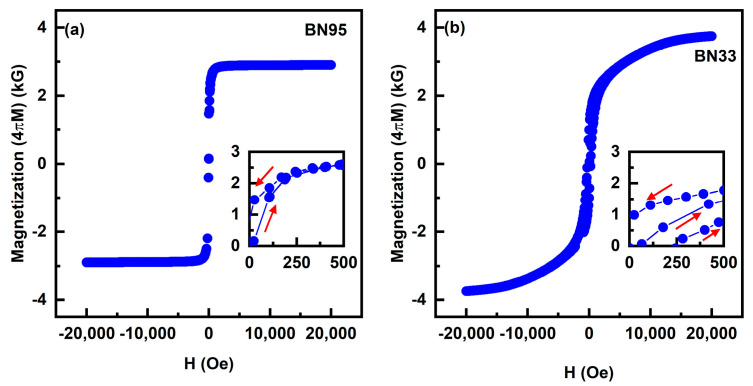
Room-temperature magnetization 4πM vs. magnetic field H data for (**a**) BN95 and (**b**) BN33 samples. The insets for low H-values clearly show the expected hysteresis and remanence in the M vs. H data. The arrows in the inset indicate which way the magnetic field increases or decreases.

**Figure 4 sensors-23-09815-f004:**
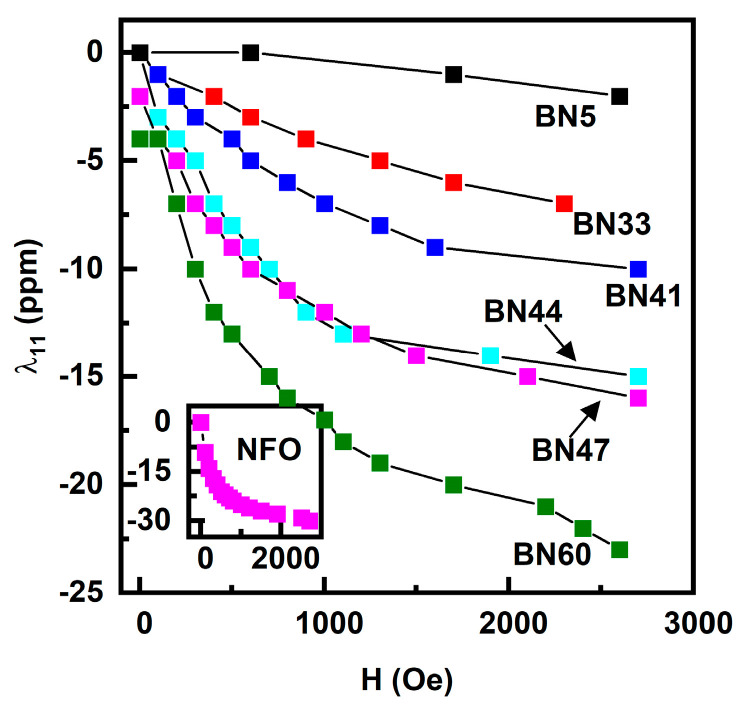
Magnetostriction, λ_11_ vs. H data measured parallel to the in-plane H for BNx composites. The magnetic field was applied parallel to the length of the sample and the strain gauge. λ_11_ for pure NFO is shown in the inset.

**Figure 5 sensors-23-09815-f005:**
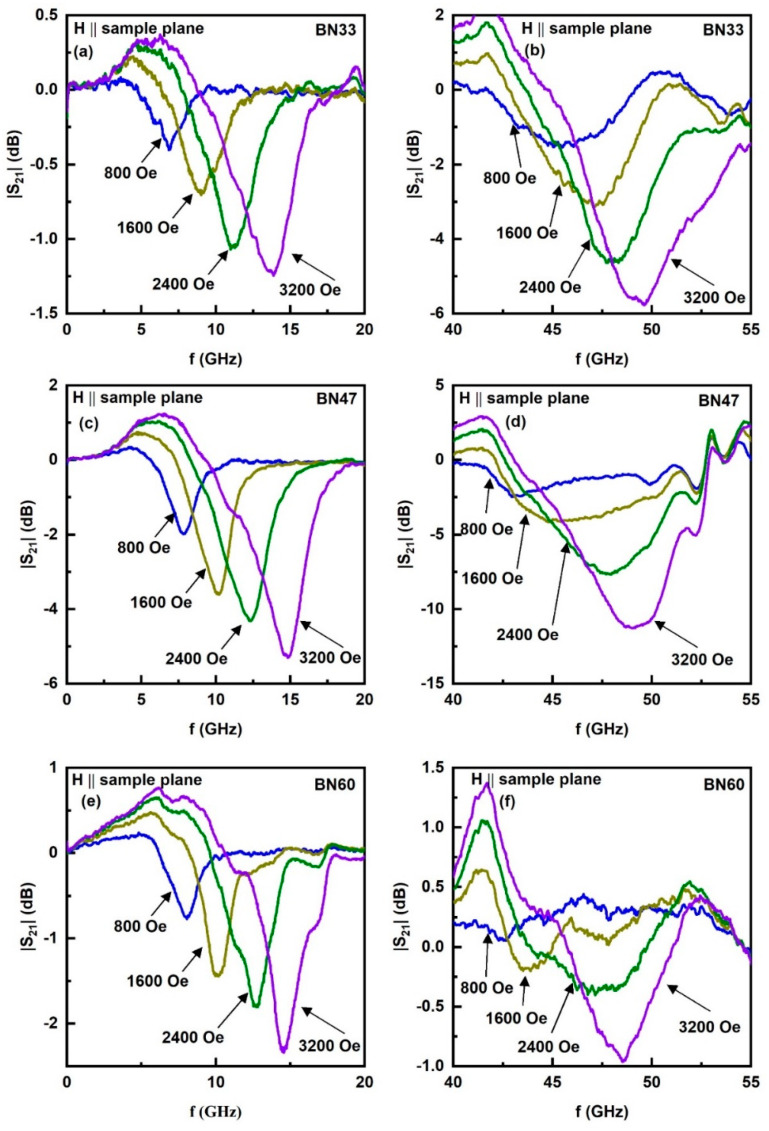
Profiles of S_21_ vs. f showing resonances in BN33, BN47, and BN60 composites for in-plane static magnetic fields. The absorption in the profiles in (**a**,**c**,**e**) for 5–20 GHz is due to ferromagnetic resonance in the NFO contents of BNx. Profiles in (**b**,**d**,**f**) show absorption due to a magneto-dielectric mode in the composites.

**Figure 6 sensors-23-09815-f006:**
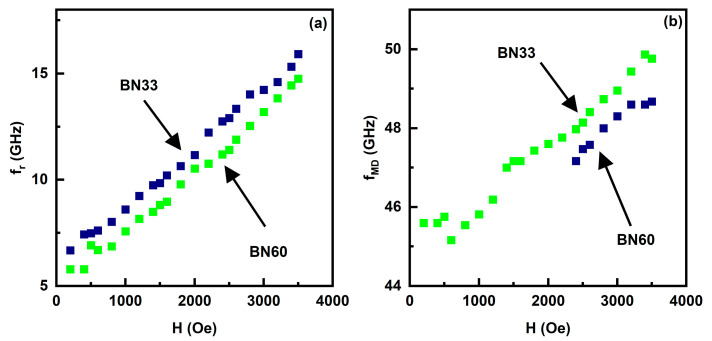
Ferromagnetic resonance frequency f_r_ as a function of H for (**a**) FMR observed in the low frequency region in Figure 5 and (**b**) magneto-dielectric mode frequency vs. H for BNx composites.

**Figure 7 sensors-23-09815-f007:**
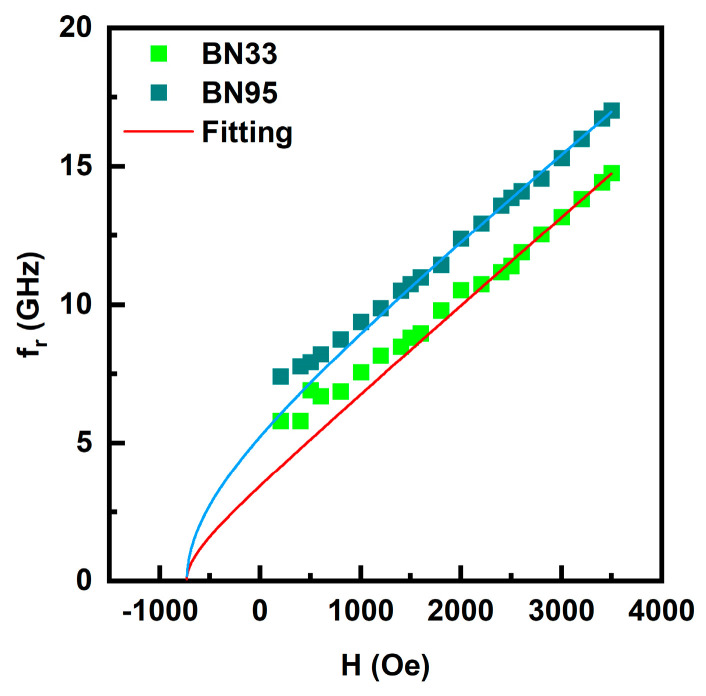
Fitting of the FMR data on f_r_ vs. H to Equation (1).

**Figure 8 sensors-23-09815-f008:**
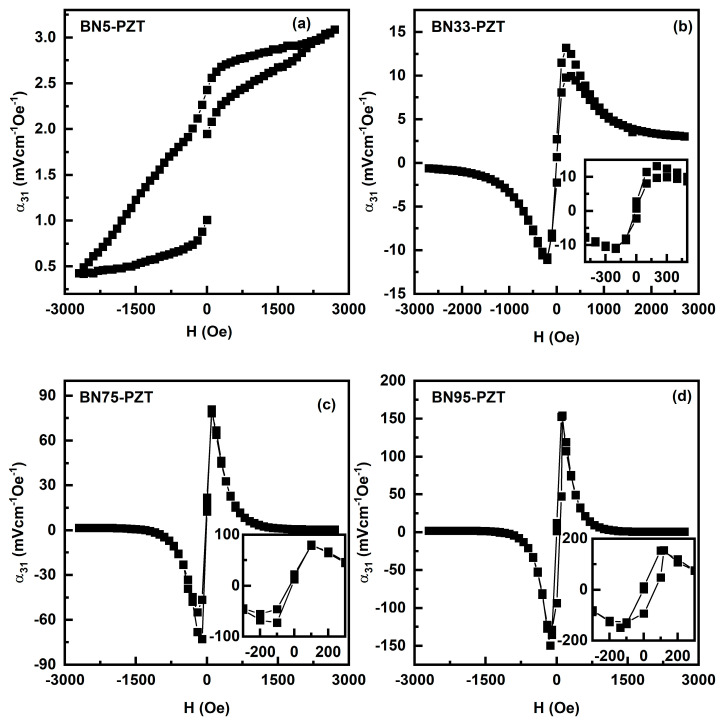
Variation of the ME voltage coefficient MEVC α_31_ with the magnetic field H for both ac field h and DC magnetic field H applied parallel to the ferrite-PZT bilayer for (**a**) BN5-PZT, (**b**) BN33-PZT, (**c**) BN75-PZT and (**d**) BN95-PZT.

**Figure 9 sensors-23-09815-f009:**
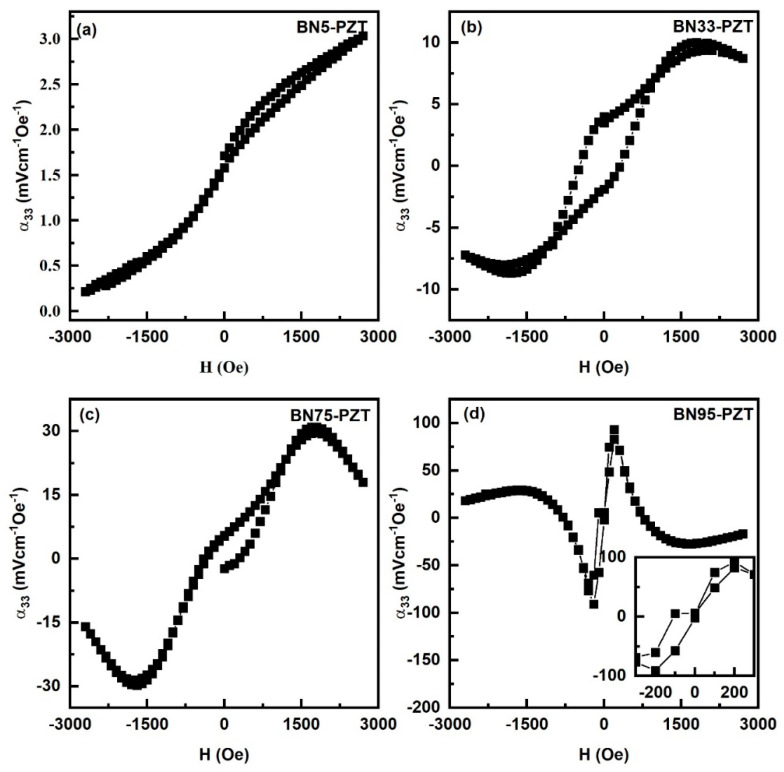
Similar MEVC α_33_ vs. H data as in Figure 7 for H, and h applied perpendicular to the sample plane for bilayers of (**a**) BN5-PZT, (**b**) BN33-PZT, (**c**) BN75-PZT and (**d**) BN95-PZT.

**Figure 10 sensors-23-09815-f010:**
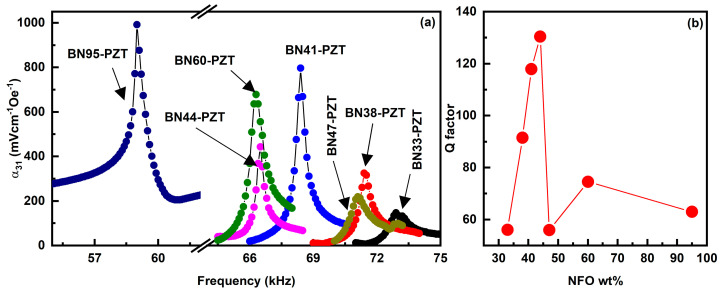
(**a**) Frequency dependence of ME coefficient α_31_ for bilayer of BNx-PZT. The peak values of MEVC occur at longitudinal mechanical resonance frequency in the samples. (**b**) Q-factor as a function of NFO weight fraction of BNx composites from the resonance ME response.

**Table 1 sensors-23-09815-t001:** Demagnetization factors for BN composites.

Sample	Demagnetization Factors		
	N_x_	N_y_	N_z_
BN33	0.11164	0.31851	0.56984
BN38	0.11238	0.32595	0.56167
BN41	0.10642	0.2778	0.61578
BN44	0.10168	0.30318	0.59514
BN47	0.11442	0.2892	0.59638
BN60	0.16197	031884	0.51919
BN75	0.19305	0.40187	0.40508
BN85	0.12413	0.22542	0.65045
BN95	0.10323	0.20369	0.69308
NFO	0.11812	0.40771	0.47417

**Table 2 sensors-23-09815-t002:** Fitting parameters for FMR and magnetic parameters for BNx composites. Gilbert damping constants are also given.

Sample	FMR Fitting Parameters		Measured Saturation Magnetization, 4πM_s_ (kG)	H_A_ (kOe)	Coercive Field (kOe)	Remanent Magnetization (M_r_) (kG)	Gilbert Damping Coefficient Calculation	
	γ (GHz/kOe)	4πM_eff_ (kOe)					Frequency Width (GHz)	Gilbert Damping Coefficient (α)
BN33	3.17	3.53	2.63	0.90	0.255	0.91	3.071	0.02450
BN38	3.26	3.46	2.68	0.78	0.161	0.77	3.042	0.02452
BN41	3.03	4.80	2.34	2.46	0.124	0.63	3.324	0.02491
BN44	2.96	5.30	2.88	2.42	0.114	0.49	3.35	0.02389
BN47	2.98	5.51	2.88	2.63	0.111	0.59	3.25	0.02354
BN60	2.61	10.07	2.75	7.32	0.089	0.73	2.739	0.01699
BN75	2.71	10.54	2.77	7.77	0.046	1.04	2.943	0.01911
BN85	2.98	7.49	2.92	4.57	0.034	0.73	2.535	0.01635
BN95	2.96	7.25	2.87	4.38	0.035	0.83	2.549	0.01613

## Data Availability

Data are available from the corresponding author upon reasonable request.

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
