# Peer review of "A Novel Spinel Ferrite-Hexagonal Ferrite Composite for Enhanced Magneto-Electric Coupling in a Bilayer with PZT"

_sensors, 2023, doi:10.3390/s23249815_

Round 1
Reviewer 1 Report
Comments and Suggestions for Authors
The authors have worked on the article A Novel Spinel Ferrite-Hexagonal Ferrite Composite for Enhanced Magneto-electric Coupling in a Bilayer with PZT. The work seems interesting; however, a few modifications and details are needed to be added before considering it for publication.
Introduction: Please elaborate on the section and add a few works in detail to demonstrate the reason for carrying out the research. Also, mention some disadvantages of previous research.
Author Response
File attached
Response to comments by Referee 1
The manuscript has been revised to address all of the comments by the four referees. Our response to specific comments by Referee 1 is given below.
Report 1
The authors have worked on the article A Novel Spinel Ferrite-Hexagonal Ferrite Composite for Enhanced Magneto-electric Coupling in a Bilayer with PZT. The work seems interesting; however, a few modifications and details are needed to be added before considering it for publication.
Introduction: Please elaborate on the section and add a few works in detail to demonstrate the reason for carrying out the research. Also, mention some disadvantages of previous research.
Response:
We thank the referee for the comment. The manuscript was revised to address this comment.
“To obtain the maximum ME voltage coefficient (αE) in a ferromagnetic-ferroelectric composite an optimized magnitude of the DC magnetic bias field H is needed. A maximum in the αE occurs when the piezomagnetic coefficient q = dλ/dH (where λ is the magnetostriction) of ferro/ferrimagnetic component of the composite is also maximum. Hence a bias magnetic field, in general, is essential to achieve a strong ME response. The need for a bias field, however, could be eliminated with a self-bias in the ferromagnetic. There are several avenues to accomplish the self-bias condition such as a large magneto-crystalline anisotropy field or the use of a functionally graded ferromagnet [7-9]. Other avenues include the use of compositionally graded ferromagnetic phase [10]. There are several experimental findings [11-13] and theoretical models [11-13] on graded ME composites. In laminated composites changing the mechanical resonance modes through electrical connectivity evoke the zero bias coupling when the bending strain activates a built-in bias [9]. Thin films that rely on magnetic field dependence of resonant frequency and angular dependence of exchange bias field [14, 15] can also show zero-bias ME effect. It is also shown that a homogeneous magnetostrictive phase can also produce zero bias ME effect [16]. Cofired layered composites consisting of textured Pb(Mg1/3Nb2/3)O3-PbTiO3 (PMN-PT) and Cu and Zn doped NiFe2O4 show a giant zero-bias ME coefficient ~1000 mV/cm Oe [17] wherein built in stress induces zero-bias effect. Very recently Huang et. al. [18] have shown very large self-bias ME effect with LiNbO3 single crystal and Ni trilayers with residual stress engineering by using Ni as electrode and piezomagnetic layer simultaneously using RF sputtering. Wu et. al. [19] have shown that large sensitivity in PMN-PT-Metglass ME sensors by utilizing shear stress induced self-bias effect. Annapureddy et. al. [20] have obtained large self-bias effect (~4200 mV/cmOe) by utilizing the graded magnetization. To date all the reported self-bias effect are sample specific. The role of the sample configuration and/or preparation is crucial to obtain the self-bias.
This work focuses on a novel, never-before used approach for a self-biased ME composite with the use of a ferromagnetic layer consisting of both M-type barium ferrite hexagonal ferrite, BaO 6Fe2O3 (BaM), with uniaxial anisotropy on the order of ~ 17.4 kOe, and nickel ferrite NiFe2O4 with high magnetostriction and piezomagnetic coefficient q [21]. Composites of BaM-NFO with (100-x) wt.% of BaM and x wt.% of NFO, (BNx), were prepared by sintering powders of both ferrites. X-ray diffraction revealed the presence of both BaM and NFO and the absence of impurity phases. Scanning electron microscopy images showed crystallites of both ferrites. Magnetization measurements at room temperature for static fields up to 2 T showed an increase in 4pM with increasing BaM content in the composite. Magnetostriction l measurements for BNx indicated an increase with increasing x and for x > 65 reached values comparable that of pure NFO. High frequency measurements were carried out to determine the anisotropy field HA from dependence of the ferromagnetic resonance (FMR) frequency fr on static magnetic field H. The in-plane HA values in BNx were found to be well above the value of 500 Oe for pure NFO. These composites show a large piezomagnetic coefficient and large magnetocrystalline anisotropy simultaneously which is somewhat unique to this spinel ferrite-hexaferrite composite.”
Reviewer 2 Report
Comments and Suggestions for Authors
The manuscript sounds well and fit for the publication. However, some modifications are required for the publication and better understanding of the readers.
1. Experimental part needs to be modified with the subsections. materials, experimental section, characterization etc...
2. Section 3: XRD: The sentence, "The XRD paterns show diffraction peaks from NFO and BaM. With increasing x-values, the NFO lines become stronger and BaM lines get weaker as expected. " Please explain, why ?
3. SEM image, comment, "BN5 also shows a similar grain distribution as pure BaM" The reason is not included.
4. what is λ11 ?
5. Section 6: Discussion must be in more details with recent and most suitable references ?
Comments on the Quality of English Languagemoderate modification
Author Response
Referees’ comments
We are grateful to the referees for the comments. The manuscript has been revised to include our response to all of the comments. Revisions to comments by Report 2 are copied below.
Report 2
The manuscript sounds well and fit for the publication. However, some modifications are required for the publication and better understanding of the readers.
- Experimental part needs to be modified with the subsections. materials, experimental section, characterization etc...
Response: We have split all of the sections on Experiments and Results into subsections as suggested.
- Section 3: XRD: The sentence, "The XRD paterns show diffraction peaks from NFO and BaM. With increasing x-values, the NFO lines become stronger and BaM lines get weaker as expected. " Please explain, why ?
Response: The XRD patterns show diffraction peaks from NFO and BaM. With increasing x-values, the NFO lines become stronger and BaM lines get weaker as expected. This is due to the reduced weight fraction of the BaM phase as we increase x. X-ray intensity corresponding to a particular phase is proportional to the weight fraction of the phase. We have deliberately varied the weight fraction of the NFO/BaM phases and when the NFO weight fraction is increased the line intensity corresponding to NFO becomes stronger, likewise BaM intensity gets weaker.
- SEM image, comment, "BN5 also shows a similar grain distribution as pure BaM" The reason is not included.
Response: When the weight fraction of the BaM phase is high (x~5) the composite is more likely to behave as pure BaM. Hence the corresponding morphological features of composites with high BaM content should also look like pure BaM. Similarly, when the weight fraction (x>41) of NFO is increased the composites tends to show a reduction in BaM grains.
- what is λ11 ?
Response: Magnetostriction of the composite on rectangular platelets were measured using a strain gage and a strain indicator/recorder (P3, Micro-Measurements). The magnetic field was applied parallel to the sample plane and along the length (direction-1) of the gage and magnetostriction measured in this configuration is labeled λ11.
- Section 6: Discussion must be in more details with recent and most suitable references ?
Response: Discussion part was revised as follows:
“It is evident from the results of this study that (i) it is possible to synthesize composites of spinel and M-type hexagonal ferrites free of ferromagnetic impurity phases, (ii) the composites, depending on the amount of BaM, have a moderately high induced planar anisotropy in all compositions, and (iii) the magnetostriction is quiet small for BaM rich composites, but increases significantly with increasing NFO content although the piezomagnetic coefficient q is rather small in all of the composites due to slow increase in l with H compared to pure NFO.
It is interesting to note that the coercive field estimated from M vs H data for the BNx composites remains well under 0.3 kOe for all compositions from x=33-95. The coercive field gradually increases from 35 Oe for BN33 to 256 Oe for BN95. BN75, the material having highest anisotropy field of 7.77 kOe have a coercive field ~45 Oe. It is also important to note that the highest remanent magnetization is also obtained for BN75 having highest anisotropy. The anisotropy acts as a driving force giving rise to a remanent magnetization for all BNx samples and a zero-bias MEVC in bilayers with PZT. All the BNx samples remains soft magnetic material with coercive field less than 300 Oe with no significant hysteresis loss for x=33-95. This value of coercive field is well below the coercive field of pure polycrystalline BaM which possesses a large coercive field ~5 kOe. In the composites the stabilization of the BaM grains (Fig. 2) does not seem to increase the coercive field.
It is also clear from the results of ME measurements in Figs.8-10 that the bilayers of BNx and PZT show MEVC that are much higher than reported values for M-type hexaferrite-PZT bilayers [37], but smaller than for NFO-PZT [38]. Under optimum value of H, the highest MEVC are α31 = 152 mV/cm Oe and α33 = 90 mV/cm Oe, both for BN95-PZT. These values, however, are relatively small due to the weak piezomagnetic coupling strengths in BNx compared to nickel ferrite or nickel zinc ferrite based layered composites with PZT [34,39].
A key and primary objective of this work was to synthesize a ferromagnetic oxide with a moderately large HA and high magnetostriction and piezomagnetic coupling for use with PZT to achieve ME coupling in the absence of an external bias magnetic field. It is worth noting that this goal was indeed accomplished. Bilayers of BNx-PZT used in this study do show a zero-bias MEVC (Figure S7 in the supplement). Bilayer of BN75-PZT shows the highest α31=22 mV/cm Oe at zero-bias and BN85-PZT bilayer shows highest α33=9 mV/cm Oe at zero-bias. It is noteworthy that the remanent magnetization of 1.04 kG for BN75 is the highest for the composites (Table 2). Strategies employed in the past to realize zero-bias ME effects included the use of an external stimuli or a functionally graded composites, either in magnetization or in composition, etc [7-17]. The use of an easy to synthesize composite of a spinel ferrite and hexaferrite in this work for zero-bias ME effect makes this method more viable than others. There are reports wherein composites consisting of NFO and PZT show large ME coefficient of 460 mV/cm Oe for bilayers and ~1200 mV/cm Oe for multilayers [40]. The MEVC at resonance in these systems was as high as ~1 V/cm Oe [41]. Modified NFO and PZT multilayers even showed a higher ME coefficient [42]. But there is hardly any evidence for ME coupling zero-bias effect in these composites [37-43].
Due to very low magnetostriction BaM is not suitable for strong direct ME coupling, but the very high uniaxial anisotropic field in the system is utilized in this work. Even though BaM grains in our BNx composites are expected to be completely randomized leading to a net zero the anisotropic field, the increase in the NFO content in BNx seems to promote the growth BaM grains with in-plane c-axis and a net in-plane anisotropy field.
Finally, we compare the zero-bias MEVC values with results reported in the past. Use of a nickel zinc ferrite graded either in magnetization or composition in a bilayer with PZT resulted in a zero-bias MEVC of 37 mV/cm Oe. Electric field induced bending vibration mode generated zero-bias ME effect in lead free system show a MEVC ~30 mV/cm-Oe [9]. Low field hysteresis based zero-bias effect also showed a value of ~60 mV/cm- Oe- [16]. In our work we have obtained zero-bias ME coefficient ~22 mV/cm-Oe- for BN75-PZT bilayer which is comparable to the earlier report [10]. The zero-bias ME response in our study could be improved with the use of composites of NFO and M-type strontium ferrite (SrM) or Al substituted SrM or BaM with higher l than pure BaM. Substituted BaM or SrM may be good choices as they also have anisotropic fields as high as ~30kG [43]. “

Reviewer 3 Report
Comments and Suggestions for Authors
Dear authors
I have some questions and comments about the article.
My questions and comments:
1. In Figure 4 we observe asymmetric resonance signals, what is the reason for this?
2. Sample BN33 shows negative Ha and sample BN95 shows positive Ha, which is difficult to notice in Figure 6, the fit drawn to all points should show Ha as the point of intersection of the fit with the H axis.
3. It would be worth determining the Gilbert damping parameter α (α is evaluated from the dependence of the linewidth ∆H on the resonance frequency) and ∆H0 (∆H0 is the inhomogeneous broadening related to layers quality).
4. Fig 4Sa, it would be necessary to divide it into two separate drawings and show how the curve with the Kittel equation was fitted.
5. In Figure 6, what does the Kittle equation mean?
6. The authors write "the higher concentration of NFO appears to promote the growth of BaM crystallites with in-plane orientation for the c-axis and a net in-plane anisotropy field that reaches a maximum value for x = 75"m is there any experimental evidence? E.g. Kikuchi diffraction?
7. To better understand the FMR results, it would be necessary to perform magnetic loop measurements using the VSM method and compare the values of anisotropy and saturation magnetization from both measurements.
I want to ask you to respond to my comments.
Author Response
Referees’ comments
We are grateful to the referees for the comments. The manuscript has been revised to include our response to all of the comments. Revisions to comments by Report 3 are copied below.
Report 3
I have some questions and comments about the article. My questions and comments:
1.In Figure 4 we observe asymmetric resonance signals, what is the reason for this?
Response: The manuscript was revised as follows:
“We utilized ferromagnetic resonance (FMR) studies in combination with the magnetization data to determine HA. Ferrite platelets, rectangular in shape, were placed in an S-shaped coplanar waveguide and excited with microwave power from a VNA. Profiles of the scattering matrix S21 as a function of frequency f were recorded. Figure 5 shows such profiles for a series of in-plane H along the sample length. For x values < 10, a single resonance mode was seen in the 50 GHz range (See Supplementary Fig. S4). With increasing NFO content two resonance modes were seen as in Fig. 5, one in the frequency range 3-20 GHz and another in the range 40-60 GHz. The S21 vs f profiles in Fig.5 (and Fig.S4) show clear asymmetry in the shape of the resonance absorption signals that can be attributed to the variations in the magnitudes of coupling between resonator and the transmission line at frequencies below and above the resonance. Such an asymmetry is not generally observed in cavity-type FMR measurements at a fixed frequency. Also, this effect is negligible for resonance modes with relatively narrow linewidth. However, for the case of transmission line broadband measurement systems and resonances with frequency-width on the order of a few GHz the asymmetry may manifest.
Another possible factor is the frequency-dependent background absorption of the coplanar line which superimposes on absorption by the resonator and leads to significant distortion of the resultant profile. Such asymmetry is most likely occur at U-band frequencies, where any imperfections of stripline, connectors, or shielding may unpredictably affect the shape of stripline transmission characteristics. The resonance frequency was estimated from frequency of maximum absorption in the profiles in Fig.5.”
- Sample BN33 shows negative Ha and sample BN95 shows positive Ha, which is difficult to notice in Figure 6, the fit drawn to all points should show Ha as the point of intersection of the fit with the H axis.
Response: The revisions to address this comment are as follows.
“The fit was not drawn through all points, since in this case the intersection with H axis will only show the magnitude of effective magnetization 4pMeff= 4pM + HA, and not HA per se. The sign and magnitude of HA can be extracted only after comparison between obtained 4pMeff and independently measured 4pM, which was done and shown in a separate figure in the revised version.
The Kittel equation in the presented form is, strictly speaking, applicable to the ferromagnetic samples of ellipsoidal shape, magnetized to saturation and with uniform static and dynamic magnetization. On the contrary, the samples presented in this investigation did not have ellipsoidal but a parallelepiped shape. That means, that even after an external magnetic field H>Nx4πM was applied and domain structure was suppressed, the sample was still not in the uniformly magnetized state. There were regions of ferrite present (mostly around edges and corners [30]) where the magnetization deviates from the direction of bias magnetic field. Thus, an even larger H should be applied before the magnetic state of the sample becomes uniform and Kittel equation becomes applicable. Due to these reasons, we took only the high-frequency (and high-field) portion of the dependencies shown in Fig. 7 for the fitting with Kittel equation since to obtain most reliable fitting parameters.
3.It would be worth determining the Gilbert damping parameter α (α is evaluated from the dependence of the linewidth ∆H on the resonance frequency) and ∆H0 (∆H0 is the inhomogeneous broadening related to layers quality).
Response: We thank the referee for the comment regarding the need for a complete understanding o the FMR parameters for the composites of NFO and BaM. We agree that in addition to the intrinsic line-width, there are contributions from several factors including inhomogeneous broadening, porosity of the polycrystalline sample, etc., In addition, the BaM phase in the composite is essentially an additional factor that will contribute to the overall FMR line-width arising from NFO. We therefore provide an approximate estimate of the damping parameter in the revised version as copied below. We do plan on a follow up rigorous study on the composites microstructure and magnetic order including FMR.
“We have also calculated the Gilbert damping coefficient (GDC) [33,34] of the composites by analyzing the FMR spectra of the samples. GDC is a dimensionless quantity which can be used as a measure of the losses in a ferromagnetic material. GDCs of the composites were calculated using the equation,
(2)
where, α is the damping coefficient, fr is the resonance frequency, ΔH is the linewidth that was estimated from the FMR frequency-width for profiles in Fig.5 and γ is the gyromagnetic ratio. Estimated values are given in Table 2. Composites with x≤47 show GDC ~0.024 and we get a smaller GDC less than 0.02 for x≥60 and is indicative of a decrease in the losses in the composites as the NFO increases. The composites seem to have a much larger damping coefficient compared to pure and dopedNFO [34,35] but smaller than the GDC of BaM [36].
(continued – Table 2 in the next page)
Table 2. Fitting parameters for FMR and magnetic parameters for BNx composites. Gilbert damping constants are also given.
|
Sample |
FMR Fitting parameters |
Measured saturation magnetization, 4πMs (kG) |
HA (kOe) |
Coercive field (kOe) |
Remanent Magnetization (Mr) (kG) |
Gilbert damping coefficient calculation |
||
|
γ (GHz/kOe) |
4πMeff (kOe) |
Frequency width (GHz) |
Gilbert damping coefficient (α) |
|||||
|
BN33 |
3.17 |
3.53 |
2.63 |
0.90 |
0.255 |
0.91 |
3.071 |
0.02450 |
|
BN38 |
3.26 |
3.46 |
2.68 |
0.78 |
0.161 |
0.77 |
3.042 |
0.02452 |
|
BN41 |
3.03 |
4.80 |
2.34 |
2.46 |
0.124 |
0.63 |
3.324 |
0.02491 |
|
BN44 |
2.96 |
5.30 |
2.88 |
2.42 |
0.114 |
0.49 |
3.35 |
0.02389 |
|
BN47 |
2.98 |
5.51 |
2.88 |
2.63 |
0.111 |
0.59 |
3.25 |
0.02354 |
|
BN60 |
2.61 |
10.07 |
2.75 |
7.32 |
0.089 |
0.73 |
2.739 |
0.01699 |
|
BN75 |
2.71 |
10.54 |
2.77 |
7.77 |
0.046 |
1.04 |
2.943 |
0.01911 |
|
BN85 |
2.98 |
7.49 |
2.92 |
4.57 |
0.034 |
0.73 |
2.535 |
0.01635 |
|
BN95 |
2.96 |
7.25 |
2.87 |
4.38 |
0.035 |
0.83 |
2.549 |
0.01613 |
4.Fig 4Sa, it would be necessary to divide it into two separate drawings and show how the curve with the Kittel equation was fitted.
Response: The figure was revised as suggested.
5.In Figure 6, what does the Kittle equation mean?
Response: We used the term to refer to the FMR resonance condition in Eq.(1).
6.The authors write "the higher concentration of NFO appears to promote the growth of BaM crystallites with in-plane orientation for the c-axis and a net in-plane anisotropy field that reaches a maximum value for x = 75"m is there any experimental evidence? E.g. Kikuchi diffraction?
Response: We agree that the above statement is only a suggestion on our part regarding the nature of the composites. We do plan on a follow up detailed study on the structure.
7.To better understand the FMR results, it would be necessary to perform magnetic loop measurements using the VSM method and compare the values of anisotropy and saturation magnetization from both measurements.
Response: We did carry out M vs H measurements suggested by the referee and used the magnetization data and the FMR estimates of the effective magnetization to estimate the anisotropy field. The revisions to address this comment are copied below.
We have carried out room-temperature measurements of the magnetization, 4πM, of the composites as a function of applied magnetic field H. Representative 4πM vs H data are shown in Fig. 3 and Fig. S3. The M vs H loops show hysteresis and remanence as expected and the saturation values of M increases with increasing BaM contents in the composites. The H-values for saturation of the magnetization is less than 3 kOe for composites for NFO rich composites and it increases with increasing the amount of BaM. The magnetization 4πM at H=20 kOe increases from 2.90 kG for BN95 to 4 kG for BN33. The highest value of remanent 4πM of 1.04 kG was measured for BN75.
Figure 3. Room-temperature magnetization 4πM vs. magnetic field H data for (a) BN95 and (b) BN33 samples. The insets for low H-values clearly show the expected hysteresis and remanence in the M vs H data.
Fig. S3. Magnetization vs. magnetic field data for BNx composites.

Reviewer 4 Report
Comments and Suggestions for Authors
In this work authors have successfully synthesized a novel ferrimagnetic composite consisting of (i) nickel ferrite with high magnetostriction and (ii) M-type barium hexaferrite with very high magneto-crystalline anisotropy field. The aim was to use such a high-q and high-HA composite to achieve strong ME coupling in the absence of a bias magnetic field in a bilayer with PZT. BNx composites with x = 5-95 wt.% had high q for NFO rich compositions and in-plane HA as high as 12 kOe for x=75. Bilayer of BN41-PZT showed a maximum MEVC ~800 mV/cm Oe at electromechanical resonance at 68.4 kHz. The use of hexaferrite-spinel ferrite composite to achieve strong zero-bias ME coupling in bilayers with PZT is significant for applications related to energy harvesting, sensors, and high frequency devices.
Remarks:
1) the graphs in Figure 5 ab are very lacking in linear approximation to establish the slope coefficient of the curve, which is discussed on page 6, lines 176 and 180.
2) The data in Figure 9 is difficult to compare with each other: different values on the Y and X coordinate axes for all compositions. The authors need to think about how to present this data in a more presentable form.
3) The hysteresis in Fig. 7a and 8a have a strong asymmetry with respect to the Y axis - the text does not indicate the reasons for this behavior of hysteresis loops.
4) From the data presented in Figure 9, it would be interesting to calculate the quality factor (Q) - have the authors thought about it? surely there is a dependence of Q from BNx.
Author Response
Response to referees’ comments:
Report 4
In this work authors have successfully synthesized a novel ferrimagnetic composite consisting of (i) nickel ferrite with high magnetostriction and (ii) M-type barium hexaferrite with very high magneto-crystalline anisotropy field. The aim was to use such a high-q and high-HA composite to achieve strong ME coupling in the absence of a bias magnetic field in a bilayer with PZT. BNx composites with x = 5-95 wt.% had high q for NFO rich compositions and in-plane HA as high as 12 kOe for x=75. Bilayer of BN41-PZT showed a maximum MEVC ~800 mV/cm Oe at electromechanical resonance at 68.4 kHz. The use of hexaferrite-spinel ferrite composite to achieve strong zero-bias ME coupling in bilayers with PZT is significant for applications related to energy harvesting, sensors, and high frequency devices.
Remarks:
1) the graphs in Figure 5 ab are very lacking in linear approximation to establish the slope coefficient of the curve, which is discussed on page 6, lines 176 and 180.
Response:
We thank the reviewer for pointing out this. We have revised the manuscript to address this comment. The FMR data in Fig.5 and Fig.6 and the supplement Fig.S5 could be fitted to resonance condition to obtain key parameters. The magneto-dielectric mode frequency vs H data for all of the composites are shown inFig.S5 and the data is somewhat scatted and not essentially linear. We have revised the manuscript as follows:
This mode in Fig.5 and Fig.S5 shows a variation in fr with H that could be approximated to a linear increase ~1.3 GHz/kOe. This slow variation in fr with H is indicative of a magneto-dielectric mode in the composite platelet. This mode is not of importance for the current study and is not considered for further analysis [27].
2) The data in Figure 9 is difficult to compare with each other: different values on the Y and X coordinate axes for all compositions. The authors need to think about how to present this data in a more presentable form.
Response:
We have changed the X and Y axis to same scales in our revised manuscript so that a proper comparison can be made. The revised Figure 10 is shown below in the next page.
Figure 10. (a) Frequency dependence of ME coefficient a31 for bilayer of BNx-PZT. The peak values of MEVC occur at longitudinal mechanical resonance frequency in the samples. (b) Q-factor as a function of NFO weight fraction of BNx composites from the resonance ME response.
3) The hysteresis in Fig. 7a and 8a have a strong asymmetry with respect to the Y axis - the text does not indicate the reasons for this behavior of hysteresis loops.
Response:
We have also noticed this asymmetry and the underlying cause is not clear. We are in the process of determining the cause and will incorporate them in our follow up work.
4) From the data presented in Figure 9, it would be interesting to calculate the quality factor (Q) - have the authors thought about it? surely there is a dependence of Q from BNx.
Response:
We have calculated the Q values for the resonance ME effects for BNx-PZT bilayers and incorporated them in Fig.10 of the revised manuscript (shown above).

Round 2
Reviewer 3 Report
Comments and Suggestions for Authors
Dear authors.
I am satisfied with the corrections made in the article.
This is a good article.
Best Regards
Reviewer 4 Report
Comments and Suggestions for Authors
The authors answered to all my questions in detail. I wish you further success!
Comments on the Quality of English Languagen/a